# Molecular Characterization of Porcine Reproductive and Respiratory Syndrome Virus in Korea from 2018 to 2022

**DOI:** 10.3390/pathogens12060757

**Published:** 2023-05-24

**Authors:** Min-A Lee, Usharani Jayaramaiah, Su-Hwa You, Eun-Gyeong Shin, Seung-Min Song, Lanjeong Ju, Seok-Jin Kang, Bang-Hun Hyun, Hyang-Sim Lee

**Affiliations:** Viral Disease Division, Animal and Plant Quarantine Agency, 177 Hyeoksin-ro, Gimcheon-si 39660, Republic of Korea; ma5147@korea.kr (M.-A.L.); ushavet85@gmail.com (U.J.); ysh0108@korea.kr (S.-H.Y.); ssing0805@naver.com (E.-G.S.); sosemi78@korea.kr (S.-M.S.); lanjeong@korea.kr (L.J.); sj.kang75@korea.kr (S.-J.K.); hyunbh@korea.kr (B.-H.H.)

**Keywords:** molecular characterization, ORF5, PRRSV, NSP2, subtype, lineage

## Abstract

Porcine reproductive and respiratory syndrome (PRRS) is an endemic disease in the Republic of Korea. Surveillance of PRRS virus (PRRSV) types is critical to tailor control measures. This study collected 5062 serum and tissue samples between 2018 and 2022. Open reading frame 5 (ORF5) sequences suggest that subgroup A (42%) was predominant, followed by lineage 1 (21%), lineage 5 (14%), lineage Korea C (LKC) (9%), lineage Korea B (LKB) (6%), and subtype 1C (5%). Highly virulent lineages 1 (NADC30/34/MN184) and 8 were also detected. These viruses typically mutate or recombine with other viruses. ORF5 and non-structural protein 2 (NSP2) deletion patterns were less variable in the PRRSV-1. Several strains belonging to PRRSV-2 showed differences in NSP2 deletion and ORF5 sequences. Similar vaccine-like isolates to the PRRSV-1 subtype 1C and PRRSV-2 lineage 5 were also found. The virus is evolving independently in the field and has eluded vaccine protection. The current vaccine that is used in Korea offers only modest or limited heterologous protection. Ongoing surveillance to identify the current virus strain in circulation is necessary to design a vaccine. A systemic immunization program with region-specific vaccinations and stringent biosecurity measures is required to reduce PRRSV infections in the Republic of Korea.

## 1. Introduction

Porcine reproductive respiratory syndrome (PRRS) is an infectious disease that contributes to massive economic losses in the swine industry. The PRRS virus (PRRSV) is responsible for reproductive failure in mature pigs and respiratory disease in pigs of all ages [1,2,3]. The virus belongs to the family *Arteriviridae* and the order *Nidovirales* and has two genotypes, PRRSV-1 (European type) and PRRSV-2 (North American type), which have been reclassified as *Betaarterivirus suid* 1 and *Betaarterivirus suid* 2, respectively [4]. They share 60% genomic identity and 20% nucleotide sequence variability within each genotype [5,6].

PRRSV is an enveloped virus with surface glycoproteins and membrane proteins embedded in lipid bilayers. The single-stranded positive-sense 15-kbp RNA is encased by the nucleocapsid. The RNA has 11 open reading frames (ORFs), as well as 5′ and 3′ untranslated regions. ORF1a and ORF1b encode non-structural proteins. ORF2-7 encode structural proteins, including nucleocapsid (N) proteins and envelope-associated proteins such as GP2a, E, GP3, GP4, GP5, and M [7,8,9,10,11].

Among the NSPs, non-structural protein 2 (NSP2) is a diverse viral protein that is important in the life cycle of PRRSV. It primarily expresses proteases involved in host immune response and viral replication. It has the highest genetic diversity and is used as a molecular marker to study the molecular epidemiology and evolution of PRRSV [12,13].

The structural protein glycoprotein 5 (GP5) is encoded by ORF5, which is crucial for the viral assembly, infectivity, and induction of neutralizing antibodies [14,15,16,17]. ORF5 exhibits considerable genetic diversity, rendering it an excellent marker for the identification of PRRSV during outbreaks and classification of the virus. 

Because of their great diversity, ORF5 (GP5) and NSP2 are frequently employed in phylogenetic and evolutionary analyses [18,19]. The ORF5 gene is considered for the global PRRSV classification system, which divides PRRSV-1 into subtypes 1–4, PRRSV-2 into lineages L1–9, and several sub-lineages [20,21]. Subtype 1 is further subdivided into subgroup A, B, and C (subtypes 1C, 1B, and 1C) [14,22,23]. This classification is used to ascertain the prevalence of existing variants and introduce new variants into any region [24]. It is possible to trace the origin of the new variant in any place by carefully tracking the import and export of animals, semen samples used for breeding, and vaccination status.

PRRSV-2 and PRRSV-1 were first discovered in South Korea in the 1980s and 2005, respectively [25,26,27,28,29]. Although PRRSV-1 and PRRSV-2 coexist in Korean swine herds, PRRSV-2 was predominant in Korea [30]; however, PRRSV-1 has spread rapidly since its first detection in 2005. The majority of PRRSV-1 is subtype 1A, with subtype 1C accounting for a minority of cases. PRRSV-2 strains found on Korean farms belong to lineages 1, 5, and 8, and nation-specific clades designated lineages Korea A (LKA), B (LKB), or C (LKC) [14,22,31].

Most of the viral population belonged to lineages 5 and 1, subtype 1A, and LKC. This virus has a high mutation and recombination rate, which has led to the emergence of new viruses in diverse fields over time. The characterization of currently circulating isolates in swine herds and on a regular basis is required to implement appropriate control strategies and prevent the emergence of new viral strains [14,27]. The genetic evolution of PRRSV strains is most frequently studied by using the ORF5 and NSP2 genes, which renders them crucial gene targets for molecular epidemiology [13,18,32]. The initial outbreak of lineage 1 occurred in North America in the early 2000s (representative strain; MN184) [33]. In 2013, a new cluster of lineage 1, with representative strain NADC30, emerged in Canada and spread to the United States and China [34]. This lineage exhibits a discontinuous 131 amino acid deletion in NSP2 and is moderately virulent; it was first isolated in the United States [35]. In 2014, a new lineage 1, with representative strain NADC34, emerged in the Unites States, and it caused large outbreaks of abortions and high mortality rates in piglets [36]. NADC34 contains a continuous deletion of 100 amino acids in NSP2. In Korea, NADC30-like viruses were first identified in 2014 [27], with NADC34-like strains being reported in 2022 [37].

PRRSV is one of the most rapidly evolving RNA viruses because of the accumulation of mutations and recombination among different lineages, which has increased the genetic diversity and complexity of PRRSV worldwide [38]. This is problematic when circulating field-virulent strains are combined with vaccine strains to produce highly pathogenic virus strains. Therefore, it is crucial to regularly monitor these recombination patterns and develop alternative strategies to stop the emergence of new strains [39,40,41,42,43]. The emergence of vaccine-like strains that are linked to previously known modified live-virus vaccine safety issues has raised concerns regarding their usage.

The objective of the current investigation was to determine the extent of the diversity exhibited by Porcine Reproductive Syndrome Virus from 2018 to 2022. We investigated the prevalence of different lineages and the genetic characteristics based on ORF5 and NSP2 proteins. 

## 2. Materials and Methods

### 2.1. Collection of Samples

A total of 5062 samples, including 2594 serum and 2468 tissue samples, was collected from different regional provinces of the Republic of Korea from 2018 to 2022 from pigs affected by reproductive failure and piglets with respiratory diseases. The tissue samples included lung tissue, lymph nodes, abortion tissue, and stillbirths. Tissue samples were homogenized by adding a medium, and the homogenized samples were centrifuged for 5 min at 5000 rpm, and the supernatant was stored at −80 °C until RNA extraction. Serum samples were used directly for subsequent experiment. 

### 2.2. RNA Extraction and PCR Amplication

Total RNA was extracted by using the Maxwell RSC viral total nucleic acid purification kit according to the manufacturer’s instructions (Promega, Madison, WI, United States). Samples were initially subjected to multiplex polymerase chain reaction (PCR) by using genotype-specific ORF7 primers and a one-step reverse transcription (RT)-PCR kit (GeNet Bio, Daejeon, Korea), which can amplify PCR products of 398 and 422 bp for PRRSV-1 and -2, respectively. The ORF7-positive samples were subjected to ORF5-based RT-PCR. The ORF5 of PRRSV was amplified by using two pairs of primers for PRRSV-1 and PRRSV-2, generating PCR products of 754 and 716 bp, respectively (Table 1). The one-step RT-PCR was carried out with the following cycling conditions: cDNA synthesis 50 °C for 30 min, pre-denaturation at 95 °C for 15 min, with 35 cycles of amplification that were performed (94 °C for 30 s, 55 °C for 30 s, 72 °C for 50 s), followed by a final extension step at 72 °C for 10 min. 

### 2.3. Phylognetic Analysis Based on ORF5 

Phylogenetic analysis was performed by using the CLC Workbench tool (QIAGEN, Aarhus A/S. Aarhus, Denmark). The phylogenetic trees were constructed via the neighbor-joining method with the Jukes–Cantor method for nucleotide distance and a bootstrap value of 1000 on ORF5 nucleotide sequence data (738 PRRSV-2 and 37 PRRSV-2 reference, 668 PRRSV-1 and 14 PRRSV-1 reference). Nucleotide and amino acid sequence identities were calculated via pairwise comparison of sequences on the CLC Workbench.

### 2.4. Analysis of Amino Acids of PRRSV

The amino acid sequences of the samples were aligned by using a CLC Workbench. The sequence logos of the samples were compared with those of the reference strains. Amino acid variations in the neutralizing epitope, B-cell epitopes, T-cell epitope, and hypervariable region were analyzed.

### 2.5. Analysis NSP2 Gene 

RT-PCR targeting *NSP2* was performed by using forward and reverse primer sets. The amplified *NSP2* gene was sequenced for further analysis. A total of 258 sequences (130 PRRSV-1 and 128 PRRSV-2) was aligned with the reference strains (three PRRSV-1 and nine PRRSV-2 references) on the CLC Workbench. Aligned sequences were extracted and translated into amino acid sequences. Aligned amino acid sequences were analyzed for deletion or insertion patterns via comparison with the reference sequences.

## 3. Results

A sum of 5062 samples was collected from farms throughout the Republic of Korea. According to data from the ORF7 analysis from 2018 to 2022 (Table 2), the prevalence of PRRSV-1, PRRSV-2, and that of both PRRSV-1 and PRRSV-2 based on ORF7 was 30% (863/2860), 59% (1689/2860), and 11% (308/2860), respectively. ORF5 RT-PCR revealed that 1847 samples tested positive for PRRSV. The overall prevalence during (2018–2022) of PRRSV-1, PRRSV-2, and PRRSV-1 and PRRSV-2 was 42% (776/1847), 52% (958/1847), and 6% (113/1847), respectively (Figure 1). Figure 2 shows the annual prevalence of PRRSV. The prevalence of PRRSV-2 was consistently higher than that of PRRSV-1 from 2018 to 2021, except in 2022, when the prevalence of PRRSV-1 was higher than that of PRRSV-2.

### 3.1. Phylogenetic Analysis

The PRRSV-1-positive samples were divided into three subgroups (A, B, and C) under subtype 1. Of the 720 sequences in the PRRSV-1 group, 89% (639) were subtype 1A, and 11% (80) were subtype 1C. Only one case of subtype 1B was reported in 2018, which was the most recent case (Figure 3A and Figure 4A). Annual prevalence data indicated that the prevalence of PRRSV-1 subtype 1A has been increasing since 2018 (Figure 3A). A total of 802 PRRSV-2 sequences was classified into 6 lineages, with 40% (322), 27% (219), 1% (9), 2% (17), 12% (95), and 17% (140) belonging to lineage 1, lineage 5, lineage 8, LKA, LKB, and LKC, respectively (Figure 3B). Lineage 1 is the most common PRRSV-2 strain, with more cases reported annually. Of the 322 positive lineage 1 samples, 40 (9.06%) branched with highly pathogenic PRRSV isolates, such as NADC30-like, NADC34-like, or MN184-like samples, and 7 samples formed a branch with lineage 8 (Figure 4B). Overall, subtype 1A (42%) was predominant, followed by lineage 1, lineage 5, LKC, LKB, and subgroup C with 21%, 14%, 9%, 6%, and 5%, respectively (Figure 3C). There were only 7 sequences of lineage 8, 15 sequences of LKA, and 1 sequence that was similar to subtype 1B.

The ORF5 nucleotide (amino acid) sequence similarity rates to PRRSV-1 vaccine strain VP-046 (subgroup C) were 90.26–99.83% (88.61–99.50%) and 79.87–88.78% (69.31–90%) for subgroup C and subgroup A. Similarly, nucleotide (amino acid) sequence similarity rates to DV (subgroup C) were 86.14–100% (84.16–100), 82.54–88.12% (68.32–88.56%) for subgroup C and subgroup A, respectively. 

Regarding sequence similarity to PRRS-2 vaccine strains, nucleotide (amino acid) sequence similarity to Ingelvac PRRSV MLV (lineage 5) was highly similar at 88.89–99.34% (85.81–100%), followed by lineage 8, LKA, LKB, lineage 1, and LKC, which were 88.23–91.71% (90.45–86.9%), 82.92–88.5% (81.82–88.07%), 77–88.02% (78.11–87.06%), 79.06–87.91% (78–87.06%), and 78.25–85.29% (78.71–86.07%), respectively (Table 3).

### 3.2. Amino Acids Analysis of PRRSV-1 Subgroup A and Subgroup C ORF5 Gene 

The ORF5 sequence logo of each subtype was compared and analyzed by using the CLC Workbench. The neutralizing epitope region of subgroup A is highly conserved, whereas subgroup C has minor V/A variation at aa position 32. Variations in amino acid sequences were observed in the signal peptide (aa positions 2, 4, 5, 8, 9, 10,16, 17, and 18); B-cell epitope 1 (aa 37); B-cell epitope 3 (aa 172, 173, and 174); T-cell epitope (aa 56, 60, 63, and 75); and T-cell epitope 2 (aa 129, 122, 123, 126, and 130) (Figure 5).

### 3.3. Analysis of Amino Acids of PRRSV-2 GP5 (ORF5 Gene) Protein

The principles of neutralizing epitopes of lineage 5, lineage 1, and LKC were implemented. LKA displayed amino acid variations in PNE at positions 38 (H/Y/N), 39 (F/S/I), and 47 (L/I) and in LKB at position 38 (N/K/T/H). Minor variations were observed in the signal peptide (aa 8, 10, 13, 14, 15, 24, 25, and 26 were conserved); T-cell epitope 1 (aa 121, 124, and 128); T-cell epitope 2 (aa 151TM1 (aa 66 and 72)); TM2 (aa 102 and 104); and B-cell epitope (aa 185, 189, 192, 199, and 200) (Figure 6).

### 3.4. Non-Structural Protein (NSP2) Deletion Pattern of PRRSV-1

NSP2 sequences of PRRSV-1 showed a typical 19 amino acid deletion pattern at aa positions 361–379, similar to the E38 (subgroup A reference strain). In addition to the standard 19 amino acid deletion, an additional 11 amino acids (in 16 sequences) from positions at aa 418–428 and 3 amino acids (in 15 sequences) at aa positions 355–357 were observed (Figure 7).

### 3.5. Non-Structural Protein (NSP2) Deletion Pattern of PRRSV-2

NSP2 sequences of PRRSV-2 were aligned and analyzed on the CLC Workbench. A total of 39 sequences did not show amino acid deletion, similar to the reference strain VR2332. In total, 4 sequences were similar to lineage 8 but had a 111 aa discontinuous deletion pattern, and 1 more sequence matched lineage 8 and showed a 6 aa deletion at aa positions 201–206. The rest of the NSP2 sequences showed a typical 131 aa discontinuous deletion pattern (111aa + 1aa + 19aa) similar to the reference strains NADC30-like, LKA, LKB, or LKC. In total, 2 of the sequences that showed 131 amino acid deletion also showed an additional amino acid deletion at aa positions 162–164, and 3 more sequences showed 5 additional amino acid deletions at aa positions 158–162 (Figure 8).

## 4. Discussion

PRRSV-1 and PRRSV-2 were found to be circulating simultaneously in Korea. The overall prevalence rates from 2018 to 2022 based on ORF5 sequences for PRRSV-1, PRRSV-2, and PRRSV-1 and PRRSV-2 were 42% (776/1847), 52% (958/1847), and 6% (113/1847), respectively. From 2018 to 2022, the overall prevalence of PRRSV-2 was higher than that of PRRSV-1, but in 2022, the prevalence of PRRSV-1 was higher than that of PRRSV-2. The percentage of mixed infection farms was also expected to increase by 2022. Previous reports show that prevalence rates of PRRSV-1 and PRRSV-2 were 29.4%, 38.4% (2007–2008), 51% and 54.4% (2013–2016), 37.4%, and 49% (2014–2019) [14,27,30]. It has been confirmed that both PRRSV-1 and PRRSV-2 coexist on swine farms in South Korea, which were similar to those in China, Hungary, and Taiwan [14,27,30,44,45,46].

The ORF5 sequences of the 1411 PRRSV isolates were analyzed and classified into several subtypes and lineages based on their phylogenetic relationships. Most PRRSV-1 samples evaluated between 2018 and 2022 belonged to subgroup A (89%). In fact, the majority of the samples analyzed, including those for PRRSV-1 and PRRSV-2, belonged to subgroup A (42%). The subgroup C population was minor, and only one subgroup B isolate tested positive for ORF5. Similar observations were made by Kim et al. [27], who found that subgroup A consistently remained in the majority of PRRSV outbreaks from 2014 to 2019.

All PRRSV-1 isolates analyzed for the NSP2 deletion pattern showed a typical 19 amino acid deletion at aa positions 361 to 379, which is similar to that of the E38 (KT033457.1) reference strain, which belongs to subgroup A. Apart from the typical 19 amino acid deletion pattern, a few isolates showed 3 amino acid deletion at aa positions 355–357, and a few others showed an 11 amino acid deletion from aa positions 418 to 428. Two amino acids, 418 and 420, are part of the ES4 epitope [47] which may lead to immune evasion and the development of distinct viral isolates. These isolates require further investigation to determine the effect of this deletion.

Subgroup A has a nucleotide similarity of 79.87–88.78% with VP-046 (UNISTRAIN PRRS, HIPRA, GIRONA, Spain), 82.54–88.12% with DV (Porcilis PRRS, MSD, the Netherlands). Subgroup C shares 86.14–100% and 90.26–99.83% similarity with the DV and VP-046 vaccine strains sequence, respectively. According to a nucleic acid analysis study, the vaccine does not completely protect animals against subgroup A infections. The PRRSV-1 vaccine used in Korean swine farms, which contains DV and or VP-046 strains, is unable to protect animals from subgroup 1A infection, and these modified live viruses revert to virulent forms, causing infection [27,48]. Antibodies exert strong positive selection pressure on PRRSV by targeting specific viral subpopulations while allowing the establishment of other subpopulations. Vaccination against the PRRSV results in genetic heterogeneity [49]. Subgroup B isolates were grouped with Thai (03RB1) and Danish strains (361–364), whereas subgroup C isolates were grouped with South Korean vaccine strains (DV MSD and VP-046 HIPRA) [50]. Immune evasion and diversifying evolution of Korean PRRSV-1 subgroup A field isolates against sequencing suggest that subgroup A is evolving independently and establishing in the Republic of Korea.

Current vaccines used in Korea contain virus strains from subgroup C, which are insufficient to protect animals affected by subgroup A infections. Therefore, a subgroup-A-specific vaccine that includes local isolates is required.

PRRSV-2 causes more severe respiratory illnesses than PRRSV-1 isolates [51]. Lineage 1 accounted for 40% (322/802) of the PRRSV-2 from 2018 to 2022. The prevalence of lineage 1 showed an increasing trend from 2018 to 2022, whereas lineage 5, LKB, and LKC showed a decreasing trend. Kim et al. [27] observed a similar pattern in their temporal dynamic study of PRRSV in Korea from 2014 to 2019. Lineage 1 is widespread in the United States and Canada. Lineage 1 may have spread owing to swine trading and artificial insemination in Korea, China, and Taiwan [27,38,46,52]. The dominance of Lineage 1 in Korea is concerning because it contains some of the most pathogenic strains, including NADC30-like, NADC34-like, and MN184 strains.

Lineage 1 was the second largest population (29.6%) of PRRSV in 2019 in Korea [27], and in Peru, 75% of the strains detected from 2015 to 2017 were NADC34-like strains [53]. According to Xu et al. [54], lineage 1 (1.5 and 1.8) accounted for 64% of positive samples in 2021. Lineage 1 strains are the most common in the Canadian provinces of Ontario and Quebec [53]. Global vaccine containing lineage 5 provided partial protection against lineage 1. It is ideal to regularly monitor PRRSV and design safe and effective vaccines that are based on current circulating strains. 

This study identified 40 highly pathogenic strains that were similar to NADC30-like, NADC34-like, and MN184 strains. Since 2005, the MN184 strain has emerged and spread throughout the Korean pig population [55]. NADC30-like strains emerged in Korea in 2015, whereas non-NADC30-like strains emerged in 2017 [23]. Since 2005, MN184-like strains have emerged and become widespread in Korean pigs [55]. The growing number of highly pathogenic strains is a major concern because they are frequently involved in recombination with other PRRSV strains. Kim et al. [23] observed lineage 1 recombinants with NADC34 as the major parent and NADC30 as the minor parent, with recombination signals in the NSP2 and NSP10 regions. According to our findings, many isolates were similar to NADC34-like strains based on the ORF5 sequence, but only three sequences were similar to those of NADC30-like virus isolates. Based on the NSP2 amino acid deletion pattern, none of the viruses resembled NADC34 in this study. However, the genomes of these isolates require further investigation. Lineage 1 isolates required recombination analysis to identify potential recombinants.

In the current study, LKB and LKC were the next most prevalent PRRSV (2018–2022). LKA, LKB, and LKC were identified in 2003, 2014 [14], and 2005, respectively [56,57]. The LKA, LKB, and LKC lineages have developed genetic components that are geographically distinct from those of the Republic of Korea, the origins of which are still unknown. ORF5 phylogenetic analysis distinguished LKA, LKB, and LKC into three distinct clusters. In contrast, whole-genome sequencing and NSP2 phylogeny merged into one large branch with two sub-branches, indicating that they share a common ancestor. 

Most Korean farms use Ingelvac PRRS modified-live vaccine (MLV) (PRRSV-2 lineage 5), which was introduced in 1995. We identified vaccine-like strains, and while the percentage of lineage 5 decreased, these vaccine-like isolates remain a concern in the field. Kim et al. [23] reported a vaccine-like strain and LKC recombination, and it was suspected that LKB was generated by the recombination of LKC and MLV strains (Ingelvac). This is supported by findings from other studies that indicate that the use of modified live viruses contributes to increased PRRSV genetic diversity [58,59,60,61]. To avoid the emergence of new viral strains or lineages, modified live viruses must be used with caution. 

Although there are fewer lineage 8 isolates, they are very important because they are classified as highly pathogenic strains [62]. In 2014, a modified live virus vaccine from Fostera PRRSV, which is an attenuated isolate of lineage 8 (P129 strain), was introduced in Korea. The origin of lineage 8 in Korean swine farms could be attributed to the introduction of the Zoetis Fostera vaccine or the importation of pigs from other countries with lineage 8 prevalence. There were seven lineage-8-like isolates based on ORF5, and four of these isolates had 111 discontinuous amino acid deletion patterns of NSP2, which is very similar to that of lineage 1 (lineage 1 shows a continuous 111 deletion pattern). Such samples require further investigation by using whole-genome sequencing.

According to Kwon et al. [31], the NSP2 amino acid deletion pattern is similar to that of the MN184-like strain, but it has a high level of nucleotide identity with VR-2332 in the ORF5–ORF7 sequence. Evidence for the possible involvement of recombination in field PRRSV evolution was first documented in the United States in 1996 and later in China [38,63,64,65]. Recombination events occur between vaccine and field strains [23,66]. Although no recombination analyses were performed in this study, it would be beneficial to examine these samples for recombination in the future.

## 5. Conclusions

From 2018 to 2022, the majority of the samples in the Republic of Korea belonged to subgroup A (42%), followed by lineages 1 (21%) and 5 (14%) and LKC (9%), LKB (6%), and subgroup C (5%). There were only a few samples from lineage 8, LKA, and subgroup B. The increase in the prevalence of subgroup A (PRRSV-1) and lineage 1 was a cause for concern. Currently available MLV vaccines contain subgroup C (PRRSV-1) and lineage 5 VR-2332 (PRRSV-2) viruses, but they do not completely protect against the more common subgroup A or lineage 1 viruses. It is well known that in case of PRRSV, there will be far less heterologous protection, indicating that the vaccine cannot protect the animals from another existing lineage. Instead, the number of infections due to subgroup A, lineage 1, LKB, and LKC increased, indicating that these isolates escaped the immunity established by vaccination with lineage 5 or subgroup C. Therefore, it is critical to develop an improved and safe vaccine that includes the prevalent type/lineage and does not mutate or recombine with field-circulating viruses. Therefore, continuous monitoring and strengthening of PRRSV prevention and control are necessary. In addition to vaccination, biosecurity measures such as restricted entry of people, supplies, and vehicles; air filtration; manure management; disposal of dead bodies; establishing PRRSV-negative boar studs and gilt sources; and careful introduction of gilts into the herd must be implemented.

## Figures and Tables

**Figure 1 pathogens-12-00757-f001:**
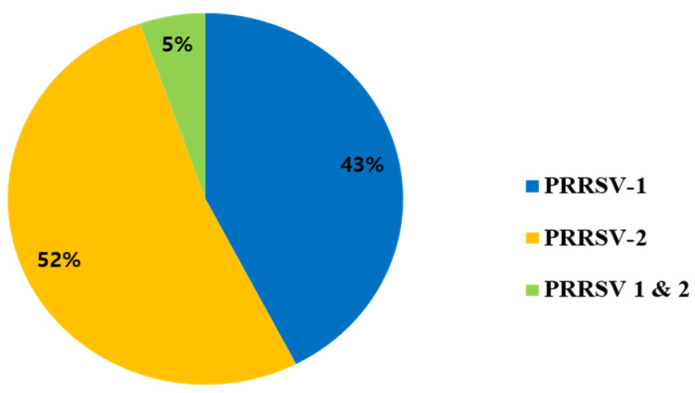
Prevalence rate of PRRSV samples (2018–2022) based on ORF5.

**Figure 2 pathogens-12-00757-f002:**
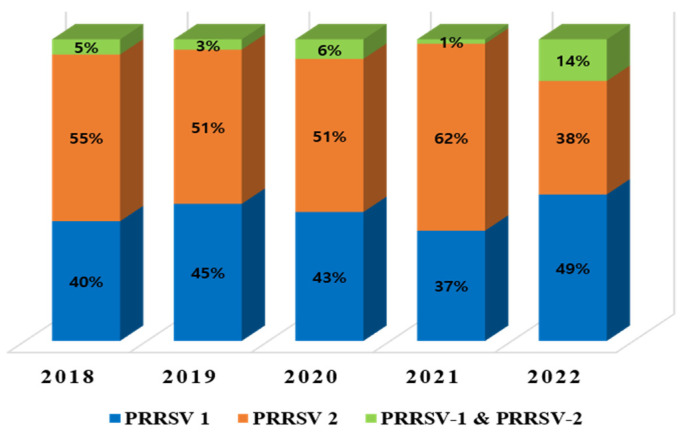
Annual prevalence of PRRSV from 2018 to 2022 based on ORF5.

**Figure 3 pathogens-12-00757-f003:**
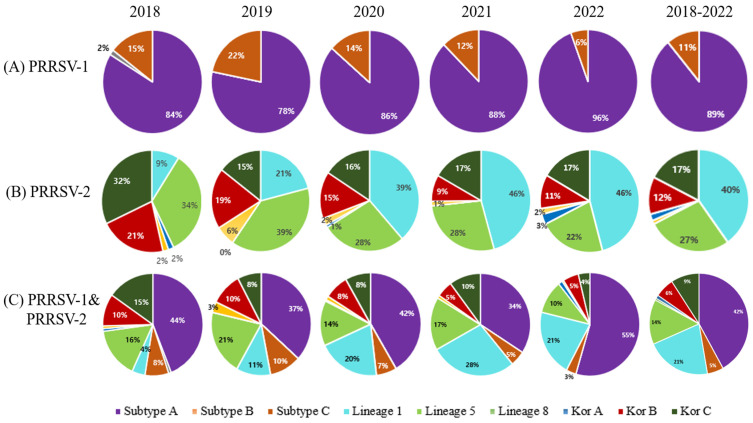
Pie charts depicting the occurrences of various subtypes and lineages of PRRSV from 2018 to 2022 in the Republic of Korea. (**A**) PRRSV-1 subgroups. (**B**) PRRSV-2 subgroups. (**C**) PRRSV-1 and PRRSV-2 subgroups/lineages.

**Figure 4 pathogens-12-00757-f004:**
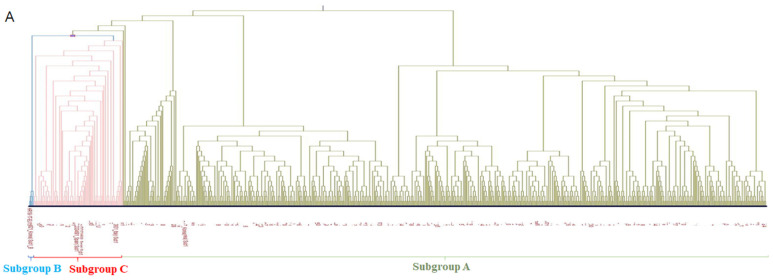
ORF 5 sequences aligned on CLC Workbench. (**A**) Phylogenetic analysis of PRRSV-1 samples for 668 PRRSV-1 samples and 14 PRRSV-1 reference strains. (**B**) Phylogenetic analysis of PRRSV-2 samples for 738 PRRSV-2 samples and 37 PRRSV reference strains. Phylogenetic tree was constructed by using neighbor-joining method and Jukes–Cantor method for nucleotide distance and a bootstrap value of 1000.

**Figure 5 pathogens-12-00757-f005:**
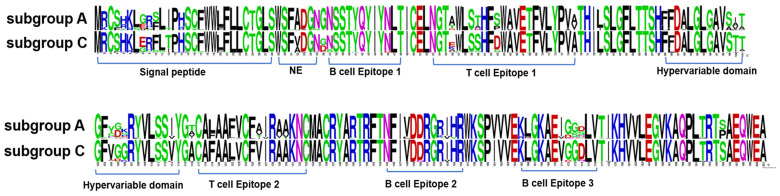
Amino acid diversity and alignment using sequence logos generated from 668 Korean PRRSV-1 GP5 sequences belonging to subgroup A and subgroup C. Amino acids were numbered based on the starting point of the GP5 domain, ranging from aa 1 to 201. The height of each amino acid letter in the sequence logos corresponds to its frequency in all the sequences from the corresponding subgroup.

**Figure 6 pathogens-12-00757-f006:**
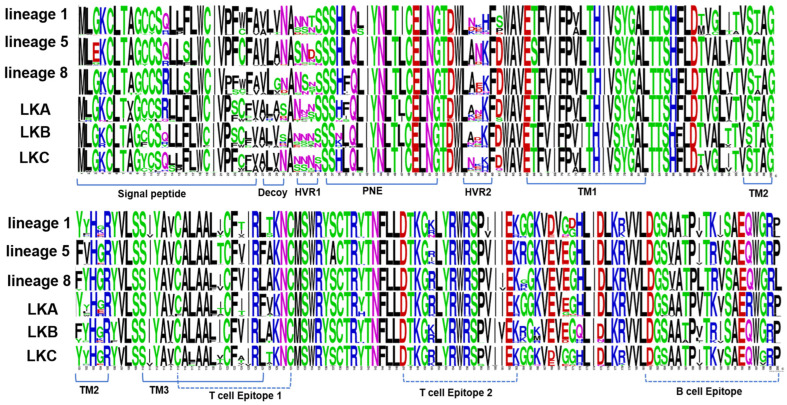
Amino acid diversity and alignment using sequence logos generated from 738 Korean PRRSV-2 GP5 sequences (aa 1–200) including all lineages and sublineages.

**Figure 7 pathogens-12-00757-f007:**
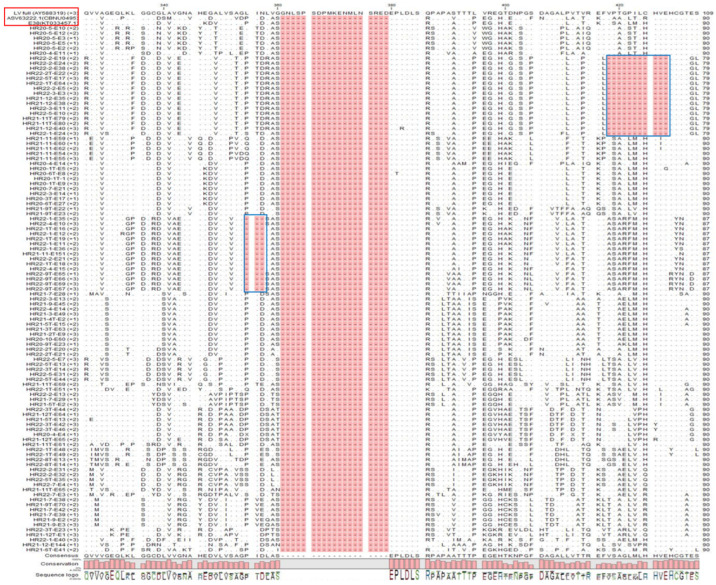
Alignment of NSP2 amino acid sequences of PRRSV-1 samples along with the reference strains (red box) to analyze deletion patterns of the amino acids by using the CLC Workbench. Standard 19 amino acid deletion and blue boxes indicated additional 11 amino acids (aa 418–428) and 3 amino acids (aa 355–357).

**Figure 8 pathogens-12-00757-f008:**
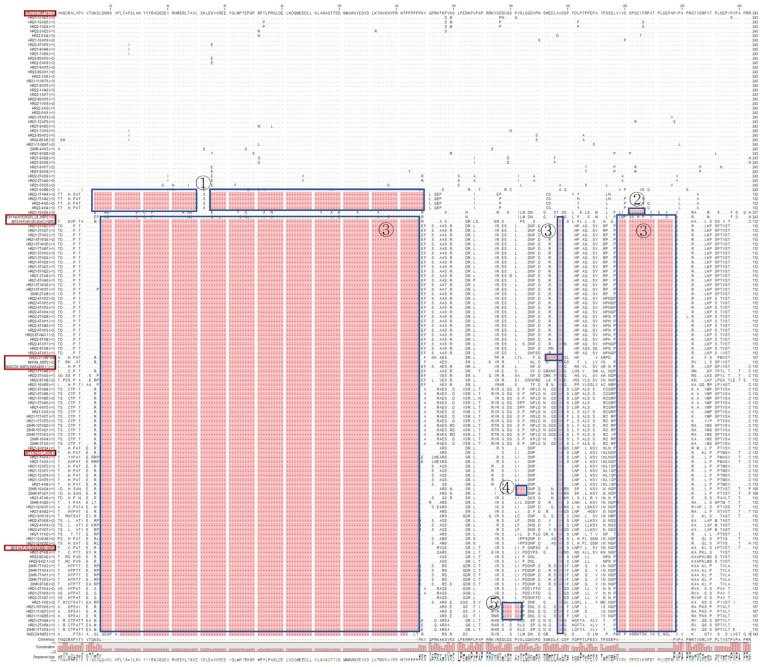
Alignment of NSP2 amino acid sequences of PRRSV-2 samples along with reference strains (red box). The majority of the samples displayed a deletion pattern resembling that of NADC30-like virus/MN184 (lineage 1) or Korea-specific clades (LKA/LKB/LKC), followed by VR2332 (lineage 5) reference strains. Very few samples showed a deletion pattern similar to that of lineage 8. 4 sequences were similar to lineage 8 but had a 111 aa discontinuous deletion pattern (blue box ①), more sequence matched lineage 8 and showed a 6 aa deletion (201–206, blue box ②). The rest of the NSP2 sequences showed a typical 131 aa discontinuous deletion pattern (111aa + 1aa + 19aa, blue box ③. 2 sequences showed 131 amino acid deletion also showed an additional amino acid deletion (162–164, blue box ④), 3 sequences showed 5 additional amino acid deletion (158–162, blue box ⑤).

**Table 1 pathogens-12-00757-t001:** Primers used for PCR amplification in the samples.

Target Gene	Genotype	Primer	Sequence (5′–3′)	Size (bp)
ORF5	PRRSV-1	EU F	5′AATGAGGTGGGCYACAACC3′	754
		EU R	5′GCGTGACACCTTAAGGGC3′	
	PRRSV-2	NA F	5′CCATTCTGGTGGCAATTTGA3′	716
		NA R	5′GGCATATATCATCACTGGCG3′	
NSP2	PRRSV-1	F	CGGCACTGTTGTYGYCCTGC	505/562
		R	AGACGCGGTGGACTTCACTG	
	PRRSV-2	F	GTGATTGAGGACTGCTGCTGTTC	907/1236
		R	GTCGATGATGGCTTGAGCTGA	

**Table 2 pathogens-12-00757-t002:** The numbers of samples analyzed from 2018 to 2022.

Year	Serum	Tissue	Total
2018	243	250	493
2019	275	270	545
2020	432	444	876
2021	1307	830	2137
2022	337	674	1011

**Table 3 pathogens-12-00757-t003:** ORF5 Nucleotide (amino acid) sequence similarity of PRRSV strain.

Genotype	Reference	Subgroup/Lineage	Sequence Similarity	
			Nucleotide (%)	Amino Acid (%)
PRRSV-1	VP-046	Subgroup C	90.26–99.83	88.61–99.50
		Subgroup A	79.87–88.78	69.31–90.0
	DV	Subgroup C	86.14–100	84.16–100
		Subgroup A	82.54–88.12	68.32–88.56
PRRSV-2	VR2332	L1	79.06–87.91	78.00–87.06
		L5	88.89–99.34	85.81–100
		L8	88.23–91.71	86.90–90.45
		LKA	82.92–88.50	81.82–88.07
		LKB	77.00–88.02	78.11–87.06
		LKC	78.25–85.29	78.71–86.07

## Data Availability

Data supporting the findings of this study are available from the corresponding author upon request.

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
