# Peer review of "Molecular Characterization of Porcine Reproductive and Respiratory Syndrome Virus in Korea from 2018 to 2022"

_pathogens, 2023, doi:10.3390/pathogens12060757_

Round 1

Reviewer 1 Report

In this manuscript, the authors described the molecular epidemiology of PRRSV in South Korea between 2018 and 2022. The following concerns have to be addressed.

1.      Please confirm if it is correct “line 53, subgroup A, B, and C (subtypes 1C, 1B, and 1C) [20,21].:

2.      Reiterate “PRRSV-2 lineage 1 has a long history its initial appearance in Canada in the 1990s.” as “PRRSV-2 lineage 1 has a long history since its initial appearance in Canada in the 1990s.”

3.      Reiterate “A total of 5,062) samples,” as “A total of 5,062 samples,”

4.      The sampling provinces should be provided, prefer to in the format of figure (geographical map is preferred)

5.      Figure 1 is missing in the manuscript.

6.      How authors explained the discrepancy of PRRSV-1, PRRSV-2, and PRRSV-1&2 positive rates by targeting ORF7 and ORF5? Which one is more reliable?

7.      What is the relationship between Korean nation-specific clades LKA, LKB, and LKC and other lineages besides 1,5, and 8 within PRRSV-2?

8.      How the authors define “highly pathogenic strains” Used in their isolates? According to the authors statement Line297-298”, the authors regarded NADC30-like, NADC34-like, ad MN184 PRRSV strains were highly pathogenic. The above statement may not be true.

The quality of English language could be improved.

Author Response

Dr. Anna Honko

Section Editor-in-Chief

Pathogens

Dear Editor:

We thank you for your time and consideration on our submission (Pathogens-2397408). Below we address the editorial office’ comments and list of changes that we made to our manuscript according to their comments. The original referee comments are provided in black color, whereas our answers are given in blue.

Reviewer 2 Report

Minor comments:

Line 30 Introduction – please be more specific, according to this sentence it seems that only piglets are affected by respiratory symptoms 

Line 120 Material and methods – why JC model was use, pretty much outdated model (from 1969…), why not some more advanced and more precise models like GTR…

Line 133 Material and methods – which algorithm was use for aligning

Figure 4. Generally, clock non-constrain tree (NJ or ML) is less suitable method to reconstruct phylogeny while presentation of results as cladogram is even less informative

Author Response

(The authors gave the same response as above.)
